# Attitudes of physicians towards target groups and content of the discharge summary: a cross-sectional analysis in Styria, Austria

Magdalena Hoffmann,[1,2,3] Christine Maria Schwarz [ID],[1,2] Gudrun Pregartner,[4] Maximilian Weinrauch,[1,2] Lydia Jantscher,[1] Lars Kamolz,[2] Gernot Brunner,[2] Gerald Sendlhofer[1,2]

For numbered affiliations see end of article.

**Correspondence to**
Christine Maria Schwarz;
christine.schwarz@medunigraz.at

## ABSTRACT

**Objectives** The discharge summary (DS) represents one of the most important instruments to ensure a safe patient discharge from the hospital. They sometimes have poor quality in content and often include medical jargon, which the patient and their relatives cannot easily understand. Therefore, many risks for patient safety exist. This study investigated the questions for whom the DS is and which contents are necessary to ensure a safe treatment.

**Design** Cross-sectional analysis.

**Setting** Styria, Austria.

**Participants** 3948 internal and external physicians were consulted.

**Interventions** An online survey consisting of 24 questions was conducted. The survey was distributed to physicians working in the province of Styria, Austria, in 2018 over a period of 6 months.

**Main outcomes and measures** Attitudes of internal and external physicians in terms of target group, content and health literacy.

**Results** In total, 1060 physicians participated in the survey. The DS is considered as a communication tool among physicians (97.9%) and the patients are also indicated as addressees (73.5%). Furthermore, there is a high level of agreement that understandable information in the DS leads to fewer questions of the patients (67.9%).

**Conclusion** In conclusion, the DS is not only seen as a document for the further treating physician but is also relevant for the patient. Incorporating the patient into their treatment at all levels may possibly strengthen the individual health literacy of the patient and their caring relatives.

## INTRODUCTION

In order to ensure a safe patient discharge from the hospital in terms of offering all relevant information, the discharge summary (DS) represents one of the most important instruments for this purpose. It usually contains diagnosis, recommendations for further therapy and treatment, monitoring appointments as well as other important information for the therapy of patients. A written DS is legally required and necessary because patients often struggle to

### Strengths and limitations of this study

► This work investigated the question for whom the discharge summary (DS) is to rethink a historical grown attitude.
► A strengths of this study is the total amount of participating physicians (n=1060).
► The DS is primarily considered as a communication tool among physicians; however, patients are also indicated as addressees.
► A limitations of the study is the moderate response rate of external physicians as well as the sample selection, which does not include private/spiritual hospitals or other regions in Austria.

accurately recall medical information given by the physician, especially when they are old or anxious. Patients tend to focus on diagnosis-related information and fail to register treatment instructions.[1] Therefore, the DS should be delivered to the patient and general practitioner (GP) to ensure a continuous treatment of the patient after discharge from hospital.[2] In practice, DS are often of poor quality in terms of information content and/or use medical jargon such as unexplained abbreviations of medical terms, which the patient and their relatives and even GP struggle to comprehend.[3 4] Choudhry et al[5] reported that for example in Rochester, USA, trauma patient DS are written at a too advanced educational level.[5] Problematic consequences for the treatment continuity can be manifold if the comprehensive medical language is n't adequately understood.

The European Health Literacy Survey report 2012 indicates that in the Austrian population the health literacy is below European Union average.[6] In addition, DS in Austria are not governed by detailed national standards for content or structure so far. Therefore, hospitals generated their own

format and selected specific contents. The legal structure of the electronic health record (ELGA), which also requires a standardisation of DS by law since 2015, is not yet widespread in Austria. The structure specifications of ELGA are only a rough guide for headings (eg, diagnosis, outcome measurement, medication, summary) in a given order; however, there are hardly any specific contents offered.[7] Furthermore, the lack of a uniform structure and of important contents in the DS is an international issue as two studies of patients with kidney disease have shown.[8 9]

Recent studies reported that patients and their relatives actively want to participate in their care and most patients want to understand their DS.[10] It is crucial that patients receive easily comprehensible written medical information as well as therapeutically relevant care information in a patient-directed DS. Lin *et al* showed in a prospective randomised controlled trial that a simple patient-directed DS delivered during a brief discussion at discharge significantly improved the patient's understanding of their illness and postdischarge recommendations, which were vice versa limited.[11] Patient-directed DS are intended to increase patient understanding as well as to improve the doctor–patient relationship.[12 13] A multitude of positive effects, such as increased patient satisfaction,[14] increased understanding, as well as a greater patient involvement in their care, were shown.[15]

The aim of this study was to investigate attitudes and perceptions among physicians in Styria regarding the current DS in terms of who the addressee of the DS is and what the necessary content is to improve the communication from hospital to the further treating GP or other healthcare workers in the future. The study was carried out by the University Hospital Graz, Austria, and involved physicians in and outside of the University hospital.

## METHODS
### Reporting
The research and reporting methodology followed by 'Recruitment process and description of the sample having access to the questionnaire'[16] recommended by equator network.

### Survey
This cross-sectional survey was based on a literature review about challenges of the medical DS[3] and contained 24 questions on specific topics (eg, target group, necessary content, contribution to health literacy, need of change) related to the DS. Each item was scored on a 4-point Likert-type scale(from 1—totally agree to 4—totally disagree) and provided the additional option of 'not relevant.' After a review process of medical professionals, the survey was pretested in 10 individuals (nurses, physicians, and staff from the quality and risk management department) several times. A detailed description of the survey is presented in table 1. The online survey was distributed via EvaSys (Electric Paper Evaluationssysteme GmbH, Germany, V.7.1) in 2018 over a period of 6 months.

| Table 1 | Content of the survey |
|---|---|
| **Item** | **Questions** |
| 2 | **The DS is a communication tool …** |
| 2.1 | … among physicians. |
| 2.2 | … for the information of patients. |
| 2.3 | … for all persons authorised by the patient (legal persons, relatives, caregivers). |
| 2.4 | The DS that is comprehensible for patients leads to less time-consuming questions. |
| 2.5 | The DS is usually not read by patients. |
| 2.6 | The patient needn't understand the DS, as it is explained by the further treating physicians. |
| 2.7 | As part of my education I have attended relevant training courses in the field of communication, such as dealing with patients and relatives. |
| 2.8 | In my education as a physician, the structure and content of the DS was an integral part. |
| 2.9 | I regularly attend training courses on communication. |
| 3 | **The following are necessary contents in the DS …** |
| 3.1 | … diagnosis. |
| 3.2 | … therapy. |
| 3.3 | … medical terminology. |
| 3.4 | … specific abbreviations. |
| 3.5 | … recommendations on further treatment. |
| 3.6 | … prescription of medication. |
| 3.7 | … control visits and follow-up appointments. |
| 3.8 | … behavioural recommendations for the patients. |
| 3.9 | … therapy recommendations/report from graduated health and nursing staff. |
| 3.10 | … therapy recommendations/report from the clinical social workers. |
| 3.11 | … therapy recommendations/report from the medical-technical services (eg, dietology, physiotherapy). |
| 3.12 | The content of the DS in its current form is sufficient for further treatment. |
| 3.13 | The DS in its present form contributes to increase the individual health literacy of patients. |
| 4.1 | In future, the DS should contain further procedures and treatment goals in an understandable way, so that patients themselves can contribute to the improvement of their health. |
| 4.2 | Should the DS be changed in its current form? |

DS, discharge summary.

### Study population
Two groups of physicians, those working in public Styrian hospitals (Styrian Hospitals Limited Liability Company (KAGes)) as well as physicians working outside as a consultant in the province of Styria, were asked to

complete the survey. The groups are referred to as 'internal' and 'external,' respectively. The internal group consisted of 2387 physicians and the external group of 1561 physicians. Both groups were invited via email to the survey. Three reminders within 4 weeks were sent to non-responders.

### Statistical analysis

The survey data were pooled into the categories 'agree' (scores 1/2) and 'disagree' (scores 3/4) and were descriptively analysed using absolute and relative frequencies. Differences between the two physician groups as well as for gender and years of work experience were determined by Fisher's exact test in an exploratory fashion and a p value of <0.05 was considered significant. When comparing more than two groups, post-hoc tests were performed and a Bonferroni correction to the significance level was applied. For three pairwise comparisons a p value <0.017 was thus considered significant. All analyses were performed using R V.3.5.3.

### Patient and public involvement

This questionnaire was developed from results of preliminary work with focus group discussions with patients, experts and stakeholders however; this present research was done without patient involvement. Our preliminary results were presented on a science to public congress for further discussion.

### RESULTS

In total, 1060 physicians participated in the survey; 747 internal physicians (response rate 31%) and 313 external physicians (response rate 20%) provided answers. For each of the 24 questions of the survey, a minimum participation rate of 88% was observed, that is, the respondent answered with one of the four intended answer categories. A detailed description of the participating physicians is presented in table 2.

### Overall survey results

Although the DS is primarily considered as a communication tool among physicians (97.9%), patients were also indicated as addressees (73.5%). Furthermore, two-thirds of the respondents indicated that easily comprehensible information leads to fewer questions from the patients (67.9%). Regarding the necessary content of the DS, diagnosis (100%), therapy (99.7%), recommendations on further treatment (99.6%), control visits and follow-up appointments (98.7%), prescription of medication (98.5%), and behavioural recommendations for the patients (94.4%) receive most agreement from the respondents. Only less than a quarter of participants (22.5%) wish to have specific abbreviations in their DS.

The DS in its current form is felt to contribute only mediocrely to health literacy (49.4%), and most respondents are of the opinion that changes should be made (46.2%). A detailed description of the physicians' ratings is presented in table 3.

### Differences between internal and external physicians

There were significant differences given by internal and external physicians for seven of the 24 items. Patients were more commonly considered as addressees of the DS by the internal group (75.8% vs 67.8%, p=0.009). The internal group was less frequently of the opinion that a DS that is comprehensible for patients leads to less

**Table 2** Baseline characteristics of the participating physicians, presented separately for internal and external physicians

| Variable | Category | Internal (n=747) | External (n=313) | Total (n=1060) |
|---|---|---|---|---|
| Gender | Female | 346 (46.3%) | 114 (36.4%) | 460 (43.4%) |
| | Male | 371 (49.7%) | 194 (62%) | 565 (53.3%) |
| | (*Missing*) | 30 (4%) | 5 (1.6%) | 35 (3.3%) |
| Work experience in years | 0–10 | 276 (36.9%) | 33 (10.5%) | 309 (29.2%) |
| | 11–20 | 201 (26.9%) | 73 (23.3%) | 274 (25.8%) |
| | >20 | 258 (34.5%) | 203 (64.9%) | 461 (43.5%) |
| | (*Missing*) | 12 (1.6%) | 4 (1.3%) | 16 (1.5%) |
| Level of work experience | Junior physicians | 37 (5%) | – | 37 (3.5%) |
| | Ward physicians | 163 (21.8%) | – | 163 (15.4%) |
| | Assistant physicians | 349 (46.7%) | – | 349 (32.9%) |
| | Senior physicians | 176 (23.6%) | – | 176 (16.6%) |
| | (*Missing*) | 22 (2.9%) | 313 (100%) | 335 (31.6%) |
| Field of work | General practitioner | – | 138 (44.1%) | 138 (13%) |
| | Specialist | – | 152 (48.6%) | 152 (14.3%) |
| | (*Missing*) | 747 (100%) | 23 (7.3%) | 770 (72.6%) |

Percentages pertain to available responses.

**Table 3** Responses to the survey items, presented separately for internal and external physicians

| Item | Category | Internal | External | Total | P value |
|---|---|---|---|---|---|
| **2. The DS is a communication tool …** | | | | | |
| 2.1 … among physicians. | 1/2 | 728 (98.1%) | 303 (97.4%) | 1031 (97.9%) | 0.483 |
| | 3/4 | 14 (1.9%) | 8 (2.6%) | 22 (2.1%) | |
| 2.2 … for the information of patients. | 1/2 | 559 (75.8%) | 208 (67.8%) | 767 (73.5%) | 0.009* |
| | 3/4 | 178 (24.2%) | 99 (32.2%) | 277 (26.5%) | |
| 2.3 … for all persons authorised by the patient (legal persons, relatives, caregivers). | 1/2 | 478 (66.5%) | 186 (61.4%) | 664 (65%) | 0.132 |
| | 3/4 | 241 (33.5%) | 117 (38.6%) | 358 (35%) | |
| 2.4 The DS that is comprehensible for patients leads to less time-consuming questions. | 1/2 | 467 (65.8%) | 224 (72.7%) | 691 (67.9%) | 0.034* |
| | 3/4 | 243 (34.2%) | 84 (27.3%) | 327 (32.1%) | |
| 2.5 The DS is usually not read by patients. | 1/2 | 315 (47.2%) | 145 (48.3%) | 460 (47.5%) | 0.781 |
| | 3/4 | 353 (52.8%) | 155 (51.7%) | 508 (52.5%) | |
| 2.6 The patient needn't understand the DS, as it is explained by the further treating physicians. | 1/2 | 173 (23.9%) | 100 (32.3%) | 273 (26.4%) | 0.006* |
| | 3/4 | 551 (76.1%) | 210 (67.7%) | 761 (73.6%) | |
| 2.7 As part of my education I have attended relevant training courses in the field of communication, such as dealing with patients and relatives. | 1/2 | 347 (48.1%) | 147 (49.7%) | 494 (48.5%) | 0.679 |
| | 3/4 | 375 (51.9%) | 149 (50.3%) | 524 (51.5%) | |
| 2.8 In my education as a physician, the structure and content of the DS was an integral part. | 1/2 | 341 (46.8%) | 171 (55.5%) | 512 (49.4%) | 0.012* |
| | 3/4 | 388 (53.2%) | 137 (44.5%) | 525 (50.6%) | |
| 2.9 I regularly attend training courses on communication. | 1/2 | 227 (31%) | 106 (34.6%) | 333 (32.1%) | 0.274 |
| | 3/4 | 506 (69%) | 200 (65.4%) | 706 (67.9%) | |
| **3. The following are necessary contents in the DS …** | | | | | |
| 3.1 … diagnosis. | 1/2 | 733 (100%) | 311 (100%) | 1044 (100%) | 1.000 |
| | 3/4 | 0 (0%) | 0 (0%) | 0 (0%) | |
| 3.2 … therapy. | 1/2 | 730 (99.6%) | 310 (100%) | 1040 (99.7%) | 0.559 |
| | 3/4 | 3 (0.4%) | 0 (0%) | 3 (0.3%) | |
| 3.3 … medical terminology. | 1/2 | 493 (68.7%) | 228 (74.5%) | 721 (70.4%) | 0.062 |
| | 3/4 | 225 (31.3%) | 78 (25.5%) | 303 (29.6%) | |
| 3.4 … specific abbreviations. | 1/2 | 162 (22.8%) | 65 (21.7%) | 227 (22.5%) | 0.742 |
| | 3/4 | 547 (77.2%) | 235 (78.3%) | 782 (77.5%) | |
| 3.5 … recommendations on further treatment. | 1/2 | 729 (99.6%) | 308 (99.7%) | 1037 (99.6%) | 1.000 |
| | 3/4 | 3 (0.4%) | 1 (0.3%) | 4 (0.4%) | |
| 3.6 … prescription of medication. | 1/2 | 720 (98.5%) | 303 (98.4%) | 1023 (98.5%) | 1.000 |
| | 3/4 | 11 (1.5%) | 5 (1.6%) | 16 (1.5%) | |
| 3.7 … control visits and follow-up appointments. | 1/2 | 721 (99%) | 303 (97.7%) | 1024 (98.7%) | 0.137 |
| | 3/4 | 7 (1%) | 7 (2.3%) | 14 (1.3%) | |
| 3.8 … behavioural recommendations for the patients. | 1/2 | 686 (95.3%) | 282 (92.5%) | 968 (94.4%) | 0.075 |
| | 3/4 | 34 (4.7%) | 23 (7.5%) | 57 (5.6%) | |
| 3.9 … therapy recommendations/report from graduated health and nursing staff. | 1/2 | 326 (46.7%) | 160 (52.8%) | 486 (48.6%) | 0.085 |
| | 3/4 | 372 (53.3%) | 143 (47.2%) | 515 (51.4%) | |
| 3.10 … therapy recommendations/report from the clinical social workers. | 1/2 | 375 (53.6%) | 171 (56.8%) | 546 (54.5%) | 0.368 |
| | 3/4 | 325 (46.4%) | 130 (43.2%) | 455 (45.5%) | |
| 3.11 … therapy recommendations/report from the medical-technical services (eg, dietology, physiotherapy). | 1/2 | 473 (67.3%) | 207 (69.2%) | 680 (67.9%) | 0.555 |
| | 3/4 | 230 (32.7%) | 92 (30.8%) | 322 (32.1%) | |
| 3.12 The content of the DS in its current form is sufficient for further treatment. | 1/2 | 571 (81%) | 216 (71.5%) | 787 (78.2%) | 0.001* |
| | 3/4 | 134 (19%) | 86 (28.5%) | 220 (21.8%) | |
| 3.13 The DS in its present form contributes to increase the individual health literacy of patients. | 1/2 | 346 (52.8%) | 115 (41.2%) | 461 (49.4%) | 0.001* |
| | 3/4 | 309 (47.2%) | 164 (58.8%) | 473 (50.6%) | |

Continued

**Table 3** Continued

| Item | Category | Internal | External | Total | P value |
|---|---|---|---|---|---|
| 4.1 In future, the DS should contain further procedures and treatment goals in an understandable way, so that patients themselves can contribute to the improvement of their health. | 1/2 | 573 (79.6%) | 240 (78.7%) | 813 (79.3%) | 0.737 |
| | 3/4 | 147 (20.4%) | 65 (21.3%) | 212 (20.7%) | |
| 4.2 Should the DS be changed in its current form? | 1/2 | 284 (41.4%) | 165 (57.9%) | 449 (46.2%) | <0.001* |
| | 3/4 | 402 (58.6%) | 120 (42.1%) | 522 (53.8%) | |

*Significant values, missing data can occur because of non-response and is not explicitly stated; percentages pertain to available responses.
DS, discharge summary.

time-consuming questions (65.8% vs 72.7%, p=0.034) and that patients need not understand the DS (23.9% vs 32.3%, p=0.006). Additionally, less internal physicians received a training in how a DS should be structured (46.8% vs 55.5%, p=0.012). Internal physicians were more often of the opinion that the content of the DS in its current form is sufficient for further treatment (81.0% vs 71.5%, p=0.001) and that the DS contributes to increase the individual health literacy of patients (52.8% vs 41.2%, p=0.001). Whereas less than half of the internal group feels the need to make changes to the current DS, more than half of the external group would like to see changes made (41.4% vs 57.9%, p<0.001). A detailed description of the physicians' ratings is presented in table 3.

### Differences according to gender
Four hundred and sixty (43.4%) of the respondents were female, 565 (53.3%) were male, and 35 (3.3%) did not disclose their gender. Female physicians more often see patients (82.2% vs 67.0%, p<0.001) and all persons authorised by the patient (70.4% vs 60.9%, p=0.002) as addressees of the DS. More of them than their male colleagues also believe that a DS that is understandable for patients leads to less time-consuming questions (71.7% vs 65.0%, p=0.029). Conversely, less female physicians agree with that the patient need not understand the DS (18.9% vs 31.8%, p<0.001) or that abbreviations are a necessary part of the DS (17.2% vs 26.0%, p=0.001). Interestingly, more female physicians want therapy recommendations or reports from other healthcare professionals, such as clinical social workers (59.5% vs 51.1%, p=0.010) or medical-technical services (72.0% vs 65.0%, p=0.022), to be included in the DS. A detailed description of the responses by gender is presented in table 4.

### Differences according to years of work experience
Three hundred and nine (29.2%) of the respondents had a work experience of up to 10 years, 274 (25.8%) had been working for 11–20 years, and 461 (43.5%) for more than 20 years; only 16 (1.5%) did not disclose this information. Physicians with up to 20 years of work experience see the patient more often as addressee of the DS than those who have been working in the field longer (80.5% of physicians with a work experience of

0–10 years vs 79.3% of those working 11–20 years vs 64% of those working >20 years, p<0.001). Physicians with less professional experience also tend to agree less with the statement that the DS does not need to be understood by patients (16.7% vs 23.5% vs 34.6%, p<0.001). The groups also showed different rates of agreement to the statement that, in future, the DS should contain further procedures and treatment goals in an understandable way, so that patients themselves can contribute to the improvement of their health (81.1% vs 84.2% vs 75.2%, p=0.011). Physicians with more work experience were more open to changes of the DS than younger ones (39.3% vs 49.0% vs 49.6%, p=0.016). A detailed description of the responses by work experience is presented in table 5.

## DISCUSSION
The main finding of this cross-sectional survey is that physicians consider themselves as the target group of the DS. The physicians have a clear view regarding the required content in the DS. However, gender and years of work experience also influence the results.

### Implications of findings
This survey showed that most physicians consider themselves as the target group of the DS; however, patients, who according to the Austrian law are the owners of the written DS,[17] are also often indicated as a target group. Patient-comprehensible DS support patients and relatives in understanding medical information as they include the most important messages when leaving the hospital. A very early study by Shankar in 2003 stated that simple and specific instructions for patients are better recalled than general statements. Patient comprehension can be aided with written or visual material.[1] Health literacy means finding, understanding, and using health information.[18] However, fewer than half of physicians said that the current DS supports health literacy of patients. A patient-directed DS could support finding the right information and understanding the offered health information.[10 19 20] Positive effects of a patient-directed DS were reported by Lin et al in 2014. Patients who received a patient-directed DS had an immediate increase in the understanding of

**Table 4** Responses to the survey items according to gender

| Item | Category | Female | Male | Total | P value |
|---|---|---|---|---|---|
| **2. The DS is a communication tool …** | | | | | |
| 2.1 … among physicians. | 1/2 | 451 (98%) | 548 (97.7%) | 1031 (97.9%) | 0.829 |
| | 3/4 | 9 (2%) | 13 (2.3%) | 22 (2.1%) | |
| 2.2 … for the information of patients. | 1/2 | 374 (82.2%) | 373 (67%) | 767 (73.5%) | <0.001* |
| | 3/4 | 81 (17.8%) | 184 (33%) | 277 (26.5%) | |
| 2.3 … for all persons authorised by the patient (legal persons, relatives, caregivers). | 1/2 | 309 (70.4%) | 336 (60.9%) | 664 (65%) | 0.002* |
| | 3/4 | 130 (29.6%) | 216 (39.1%) | 358 (35%) | |
| 2.4 The DS that is comprehensible for patients leads to less time-consuming questions. | 1/2 | 321 (71.7%) | 353 (65%) | 691 (67.9%) | 0.029* |
| | 3/4 | 127 (28.3%) | 190 (35%) | 327 (32.1%) | |
| 2.5 The DS is usually not read by patients. | 1/2 | 194 (46.3%) | 254 (48.8%) | 460 (47.5%) | 0.470 |
| | 3/4 | 225 (53.7%) | 267 (51.2%) | 508 (52.5%) | |
| 2.6 The patient needn't understand the DS, as it is explained by the further treating physicians. | 1/2 | 86 (18.9%) | 174 (31.8%) | 273 (26.4%) | <0.001* |
| | 3/4 | 368 (81.1%) | 374 (68.2%) | 761 (73.6%) | |
| 2.7 As part of my education I have attended relevant training courses in the field of communication, such as dealing with patients and relatives. | 1/2 | 207 (46.8%) | 273 (50.1%) | 494 (48.5%) | 0.337 |
| | 3/4 | 235 (53.2%) | 272 (49.9%) | 524 (51.5%) | |
| 2.8 In my education as a physician, the structure and content of the DS was an integral part. | 1/2 | 209 (46.4%) | 286 (51.5%) | 512 (49.4%) | 0.113 |
| | 3/4 | 241 (53.6%) | 269 (48.5%) | 525 (50.6%) | |
| 2.9 I regularly attend training courses on communication. | 1/2 | 142 (31.1%) | 179 (32.5%) | 333 (32.1%) | 0.684 |
| | 3/4 | 314 (68.9%) | 372 (67.5%) | 706 (67.9%) | |
| **3. The following are necessary contents in the DS …** | | | | | |
| 3.1 … diagnosis. | 1/2 | 455 (100%) | 557 (100%) | 1044 (100%) | 1.000 |
| | 3/4 | 0 (0%) | 0 (0%) | 0 (0%) | |
| 3.2 … therapy. | 1/2 | 453 (99.8%) | 555 (99.6%) | 1040 (99.7%) | 1.000 |
| | 3/4 | 1 (0.2%) | 2 (0.4%) | 3 (0.3%) | |
| 3.3 … medical terminology. | 1/2 | 310 (69.5%) | 387 (70.7%) | 721 (70.4%) | 0.676 |
| | 3/4 | 136 (30.5%) | 160 (29.3%) | 303 (29.6%) | |
| 3.4 … specific abbreviations. | 1/2 | 76 (17.2%) | 140 (26%) | 227 (22.5%) | 0.001* |
| | 3/4 | 365 (82.8%) | 398 (74%) | 782 (77.5%) | |
| 3.5 … recommendations on further treatment. | 1/2 | 450 (99.6%) | 555 (99.6%) | 1037 (99.6%) | 1.000 |
| | 3/4 | 2 (0.4%) | 2 (0.4%) | 4 (0.4%) | |
| 3.6 … prescription of medication. | 1/2 | 448 (98.7%) | 543 (98.2%) | 1023 (98.5%) | 0.619 |
| | 3/4 | 6 (1.3%) | 10 (1.8%) | 16 (1.5%) | |
| 3.7 … control visits and follow-up appointments. | 1/2 | 447 (98.7%) | 547 (98.7%) | 1024 (98.7%) | 1.000 |
| | 3/4 | 6 (1.3%) | 7 (1.3%) | 14 (1.3%) | |
| 3.8 … behavioural recommendations for the patients. | 1/2 | 420 (94.2%) | 517 (94.5%) | 968 (94.4%) | 0.890 |
| | 3/4 | 26 (5.8%) | 30 (5.5%) | 57 (5.6%) | |
| 3.9 … therapy recommendations/report from graduated health and nursing staff. | 1/2 | 218 (50.5%) | 252 (46.7%) | 486 (48.6%) | 0.246 |
| | 3/4 | 214 (49.5%) | 288 (53.3%) | 515 (51.4%) | |
| 3.10 … therapy recommendations/report from the clinical social workers. | 1/2 | 260 (59.5%) | 274 (51.1%) | 546 (54.5%) | 0.010* |
| | 3/4 | 177 (40.5%) | 262 (48.9%) | 455 (45.5%) | |
| 3.11 … therapy recommendations/report from the medical-technical services (eg, dietology, physiotherapy). | 1/2 | 314 (72%) | 349 (65%) | 680 (67.9%) | 0.022* |
| | 3/4 | 122 (28%) | 188 (35%) | 322 (32.1%) | |
| 3.12 The content of the DS in its current form is sufficient for further treatment. | 1/2 | 333 (76.2%) | 431 (80%) | 787 (78.2%) | 0.161 |
| | 3/4 | 104 (23.8%) | 108 (20%) | 220 (21.8%) | |

**Table 4** Continued

| Item | Category | Female | Male | Total | P value |
|---|---|---|---|---|---|
| 3.13 The DS in its present form contributes to increase the individual health literacy of patients. | 1/2 | 192 (47.9%) | 252 (49.9%) | 461 (49.4%) | 0.548 |
| | 3/4 | 209 (52.1%) | 253 (50.1%) | 473 (50.6%) | |
| 4.1 In future, the DS should contain further procedures and treatment goals in an understandable way, so that patients themselves can contribute to the improvement of their health. | 1/2 | 364 (81.6%) | 426 (77.9%) | 813 (79.3%) | 0.155 |
| | 3/4 | 82 (18.4%) | 121 (22.1%) | 212 (20.7%) | |
| 4.2 Should the DS be changed in its current form? | 1/2 | 198 (46%) | 238 (46.3%) | 449 (46.2%) | 0.948 |
| | 3/4 | 232 (54%) | 276 (53.7%) | 522 (53.8%) | |

*Significant values, missing data can occur because of non-response and is not explicitly stated; percentages pertain to available responses.
DS, discharge summary.

tests performed at the hospital as well as a higher compliance in post-discharge recommendations.[11] Wernick *et al*[21] reported in 2016 that minimising the use of medical terminology in medical correspondence significantly improved patient understanding and perception of their ability to manage chronic health conditions.[16] Our results also showed that the majority of physicians consider medical terminology a necessary part of DS but disapprove the use of abbreviations.

Our results show that physicians have a clear expectation regarding the required content in the DS. The need for specific content in Styria is similar to other studies.[22] The use of ELGA criteria, providing a minimum number of (mandatory and optional) headings in a fixed structure, to guide physicians in writing a comprehensive DS holds promise in improving the existing deficits in communication at transitions of care, as another study using a standardised DS already showed.[14]

For a majority of physicians, abbreviations are undesirable since they are not only problematic for patients but also for physicians. Chemali *et al*[23] reported in 2015 that abbreviations used in hospital DS are not well understood by the GPs who receive them.[17] In another study of Shilo and Shilo, abbreviations from 80 DS and admission notes from orthopaedic surgery and medical wards were extracted and graded by senior physicians as understandable, ambiguous, or unknown.[24] To improve comprehension of the DS the use of (ambiguous and unknown) abbreviations should be decreased.

It is noteworthy that female physicians expressed that the DS should contain more reports from other healthcare workers. Furthermore, they see a greater need for patients to understand the DS than male physicians do. The differences in answers related to the years of work experience may indicate a paradigm shift. Physicians with fewer years of experience see the patient significantly more often as addressee of the DS compared with physicians with more than 20 years of work experience. The authors are not aware of comparable findings from current literature.

Our results showed that less than half of the physicians had received training to write the DS. However, physicians with more years of work experience more frequently stated to have received specific training as part of their education. In a study by Yemm *et al*[25] in 2014, 36% of young physicians reported that they had inadequate training in composing a DS.[15] Therefore, it may be an advantage for young medical professionals to have a fixed structure and guidelines for its content. However, a good structure and layout does not replace the training in writing medical DS. A study by Shivji *et al* has demonstrated that educational sessions improved the quality of written DS.[26] We suggest that a course in writing DS should be included as part of the curriculum in medical schools, as it is important for junior doctors to learn how to prepare a DS.

Furthermore, to improve the DS, supporting instruments and standards need to be implemented. In 2018 Savvopoulos *et al*[27] developed a scoring tool to assess the quality of the DS.[16] Such tools can help to assess and improve the quality of current DS and derive any need for improvement if necessary.

### Strengths and limitations of the study

A strength of this study is the high response rate of internal physicians and the involvement of occupational groups and patients in the project process before and after the survey.

This study has several limitations. This survey is a snapshot of current attitudes of doctors regarding discharge information. Another limitation are the dropped non-responders of the survey. A limitation of the present study is the sample selection, which does not include private/spiritual hospitals or other regions in Austria. Another important limitation of our study is the moderate response rate of external physicians that constitutes a potential bias. However, this phenomenon was also observed in other surveys with externals. It might be that external physicians can hardly be attracted for participation.

### CONCLUSIONS

Based on our results, it can be concluded that the DS is perceived not only as a document for the further treating

**Table 5** Responses to the survey items according to years of work experience

| Item | Category | 0–10 | 11–20 | >20 | Total | P value |
|------|----------|------|-------|-----|-------|---------|
| **2. The DS is a communication tool …** | | | | | | |
| 2.1 … among physicians. | 1/2 | 303 (98.4%) | 261 (96%) | 454 (98.7%) | 1031 (97.9%) | 0.050* |
| | 3/4 | 5 (1.6%) | 11 (4%) | 6 (1.3%) | 22 (2.1%) | |
| 2.2 … for the information of patients. | 1/2 | 248 (80.5%) | 214 (79.3%) | 293 (64.8%) | 767 (73.5%) | <0.001†‡ |
| | 3/4 | 60 (19.5%) | 56 (20.7%) | 159 (35.2%) | 277 (26.5%) | |
| 2.3 … for all persons authorised by the patient (legal persons, relatives, caregivers). | 1/2 | 201 (68.4%) | 188 (69.4%) | 264 (59.5%) | 664 (65%) | 0.008†‡ |
| | 3/4 | 93 (31.6%) | 83 (30.6%) | 180 (40.5%) | 358 (35%) | |
| 2.4 The DS that is comprehensible for patients leads to less time-consuming questions. | 1/2 | 207 (69.2%) | 193 (73.1%) | 281 (63.6%) | 691 (67.9%) | 0.026‡ |
| | 3/4 | 92 (30.8%) | 71 (26.9%) | 161 (36.4%) | 327 (32.1%) | |
| 2.5 The DS is usually not read by patients. | 1/2 | 157 (56.3%) | 119 (47.6%) | 178 (41.9%) | 460 (47.5%) | 0.001† |
| | 3/4 | 122 (43.7%) | 131 (52.4%) | 247 (58.1%) | 508 (52.5%) | |
| 2.6 The patient needn't understand the DS, as it is explained by the further treating physicians. | 1/2 | 51 (16.7%) | 62 (23.5%) | 156 (34.6%) | 273 (26.4%) | <0.001†‡ |
| | 3/4 | 254 (83.3%) | 202 (76.5%) | 295 (65.4%) | 761 (73.6%) | |
| 2.7 As part of my education I have attended relevant training courses in the field of communication, such as dealing with patients and relatives. | 1/2 | 151 (50.3%) | 115 (43.7%) | 222 (50.1%) | 494 (48.5%) | 0.196 |
| | 3/4 | 149 (49.7%) | 148 (56.3%) | 221 (49.9%) | 524 (51.5%) | |
| 2.8 In my education as a physician, the structure and content of the DS was an integral part. | 1/2 | 128 (42.5%) | 137 (50.7%) | 241 (53.2%) | 512 (49.4%) | 0.014† |
| | 3/4 | 173 (57.5%) | 133 (49.3%) | 212 (46.8%) | 525 (50.6%) | |
| 2.9 I regularly attend training courses on communication. | 1/2 | 74 (24.3%) | 88 (32.8%) | 167 (36.9%) | 333 (32.1%) | 0.001†‡ |
| | 3/4 | 230 (75.7%) | 180 (67.2%) | 286 (63.1%) | 706 (67.9%) | |
| **3. The following are necessary contents in the DS …** | | | | | | |
| 3.1 … diagnosis. | 1/2 | 304 (100%) | 273 (100%) | 454 (100%) | 1044 (100%) | 1.000 |
| | 3/4 | 0 (0%) | 0 (0%) | 0 (0%) | 0 (0%) | |
| 3.2 … therapy. | 1/2 | 303 (99.7%) | 271 (100%) | 453 (99.6%) | 1040 (99.7%) | 0.794 |
| | 3/4 | 1 (0.3%) | 0 (0%) | 2 (0.4%) | 3 (0.3%) | |
| 3.3 … medical terminology. | 1/2 | 196 (64.9%) | 194 (72.4%) | 321 (72.6%) | 721 (70.4%) | 0.055 |
| | 3/4 | 106 (35.1%) | 74 (27.6%) | 121 (27.4%) | 303 (29.6%) | |
| 3.4 … specific abbreviations. | 1/2 | 58 (19.6%) | 64 (24.1%) | 103 (23.6%) | 227 (22.5%) | 0.347 |
| | 3/4 | 238 (80.4%) | 202 (75.9%) | 334 (76.4%) | 782 (77.5%) | |
| 3.5 … recommendations on further treatment. | 1/2 | 303 (99.7%) | 269 (99.3%) | 453 (99.8%) | 1037 (99.6%) | 0.696 |
| | 3/4 | 1 (0.3%) | 2 (0.7%) | 1 (0.2%) | 4 (0.4%) | |
| 3.6 … prescription of medication. | 1/2 | 300 (99%) | 266 (98.2%) | 444 (98.2%) | 1023 (98.5%) | 0.655 |
| | 3/4 | 3 (1%) | 5 (1.8%) | 8 (1.8%) | 16 (1.5%) | |
| 3.7 … control visits and follow-up appointments. | 1/2 | 299 (99.7%) | 270 (99.3%) | 442 (97.6%) | 1024 (98.7%) | 0.041* |
| | 3/4 | 1 (0.3%) | 2 (0.7%) | 11 (2.4%) | 14 (1.3%) | |
| 3.8 … behavioural recommendations for the patients. | 1/2 | 290 (97.3%) | 254 (94.1%) | 411 (92.6%) | 968 (94.4%) | 0.016†‡ |
| | 3/4 | 8 (2.7%) | 16 (5.9%) | 33 (7.4%) | 57 (5.6%) | |
| 3.9 … therapy recommendations/report from graduated health and nursing staff. | 1/2 | 133 (45.2%) | 124 (48.1%) | 221 (50.6%) | 486 (48.6%) | 0.364 |
| | 3/4 | 161 (54.8%) | 134 (51.9%) | 216 (49.4%) | 515 (51.4%) | |
| 3.10 … therapy recommendations/report from the clinical social workers. | 1/2 | 166 (56.3%) | 134 (51.3%) | 239 (55.1%) | 546 (54.5%) | 0.476 |
| | 3/4 | 129 (43.7%) | 127 (48.7%) | 195 (44.9%) | 455 (45.5%) | |
| 3.11 … therapy recommendations/report from the medical-technical services (eg, dietology, physiotherapy). | 1/2 | 201 (67.9%) | 168 (65.1%) | 302 (69.1%) | 680 (67.9%) | 0.552 |
| | 3/4 | 95 (32.1%) | 90 (34.9%) | 135 (30.9%) | 322 (32.1%) | |
| 3.12 The content of the DS in its current form is sufficient for further treatment. | 1/2 | 234 (79.6%) | 203 (78.1%) | 340 (77.1%) | 787 (78.2%) | 0.724 |
| | 3/4 | 60 (20.4%) | 57 (21.9%) | 101 (22.9%) | 220 (21.8%) | |
| 3.13 The DS in its present form contributes to increase the individual health literacy of patients. | 1/2 | 145 (52.3%) | 129 (52.7%) | 182 (45.3%) | 461 (49.4%) | 0.095 |
| | 3/4 | 132 (47.7%) | 116 (47.3%) | 220 (54.7%) | 473 (50.6%) | |

Continued

**Table 5**  Continued

| Item | Category | 0–10 | 11–20 | >20 | Total | P value |
|------|----------|------|-------|-----|-------|---------|
| 4.1 In future, the DS should contain further procedures and treatment goals in an understandable way, so that patients themselves can contribute to the improvement of their health. | 1/2 | 245 (81.1%) | 224 (84.2%) | 333 (75.2%) | 813 (79.3%) | 0.011‡ |
| | 3/4 | 57 (18.9%) | 42 (15.8%) | 110 (24.8%) | 212 (20.7%) | |
| 4.2 Should the DS be changed in its current form? | 1/2 | 114 (39.3%) | 121 (49%) | 209 (49.6%) | 449 (46.2%) | 0.016† |
| | 3/4 | 176 (60.7%) | 126 (51%) | 212 (50.4%) | 522 (53.8%) | |

Missing data can occur because of non-response and is not explicitly stated; percentages pertain to available responses.
*No pairwise difference remained significant after Bonferroni correction.
†Significant differences (p<0.017) between 0–10 and >20 years.
‡Significant differences (p<0.017) between 11–20 and >20 years.
DS, discharge summary.

physician but also for the patient and other caregivers. Now, the DS primarily is written for the further treating doctor and has often not a uniform structure. The question arises how a DS can be designed, that is partly understandable for the patient and relatives and which can be implemented in the clinic practice. Further research on this topic should address this question. An international comparison of physicians' perceptions, attitudes and solutions regarding a patient-directed DS, important contents, and improvements in health literacy of patients would be beneficial.

**Author affiliations**
[1]Executive Department for Quality and Risk Management, Hospital of the Federal State of Styria and University Hospital Graz, Graz, Austria
[2]Research Unit for Safety in Health, Division of Plastic, Aesthetic and Reconstructive Surgery, Department of Surgery, Medical University of Graz, Graz, Austria
[3]Division of Endocrinology and Diabetology, Department of Internal Medicine, Medical University of Graz, Graz, Austria
[4]Institute for Medical Informatics, Statistics und Documentation, Medical University of Graz, Graz, Austria

**Contributors**  Guarantor: MH. Study concept and design: MH, GB, MW, GS. Acquisition of data: MH, CMS, LJ, LK. Analysis and interpretation of pooled data: MH, CMS, GP, LJ, LK. Drafting of the manuscript: MH, CMS, GS. Critical revision of the manuscript for important intellectual content: All authors. Statistical analysis of pooled data: GP, LJ.

**Funding**  This work was supported by the Healthcare fund Styria (Gesundheitsfonds Steiermark).

**Competing interests**  None declared.

**Patient consent for publication**  Not required.

**Ethics approval**  The Ethics Committee of the Medical University of Graz approved the study (vote#: 30-220 ex17/18).

**Provenance and peer review**  Not commissioned; externally peer reviewed.

**Data availability statement**  Data are available upon reasonable request.

**ORCID iD**
Christine Maria Schwarz http://orcid.org/0000-0003-0924-2340

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
