## [Reviewer comments · BMJ Open]

ARTICLE DETAILS

TITLE (PROVISIONAL)	Attitudes of physicians towards target groups and content of the discharge summary– a cross-sectional analysis in Styria, Austria
AUTHORS	Hoffmann, Magdalena; Schwarz, Christine; Pregartner, Gudrun; Weinrauch, Maximilian; Jantscher, Lydia; Kamolz, Lars; Brunner, Gernot; Sendlhofer, Gerald

VERSION 1 – REVIEW

REVIEWER	Mario Iannaccone S.G. Bosco Hospital, Turin, Italy
REVIEW RETURNED	23-Oct-2019

GENERAL COMMENTS	The authors should be congratulated for their efforts, the paper is interesting and original. I have some minor comment The authors should report in the limitation section the low response rate of the physicians as a potential bias The authors should comment about the 88% of participation rate, if it was pre-specified or not A General practitioner vs specialist analysis could be of interest
--

REVIEWER	Zhixin Liu Stats Central, UNSW Sydney, Australia
REVIEW RETURNED	04-Nov-2019

GENERAL COMMENTS	1. Statistical analysis: is there any particular reason the authors chose Fisher Exact test? Given the large sample size, Chi-square test could be more appropriate. 2. Result: (1) Would be interested to see comparison between the internal and external group in key characteristic of physicians, e.g. age, gender, experience etc. (2) Working experience are classified into three groups, seems the p-value reported are the overall p-value for the comparison across three groups. Further pairwise examination would help to identify where the association with the outcome is.
---

VERSION 1 – AUTHOR RESPONSE

Reviewer(s)' Comments to Author:

Reviewer: 1

Reviewer Name: Mario Iannaccone

Institution and Country: S.G. Bosco Hospital, Turin, Italy

Please state any competing interests or state 'None declared': none declared

Please leave your comments for the authors below

The authors should be congratulated for their efforts, the paper is interesting and original.

Thank you very much!

I have some minor comment

The authors should report in the limitation section the low response rate of the physicians as a potential bias

Thank you for your suggestion: We have added a sentence to state the potential bias of the low response rate of the physicians.

“Another important limitation of our study is the moderate response rate of external physicians that constitutes a potential bias. However, this phenomenon was also observed in other surveys with externals. It might be that external physicians can hardly be attracted for participation”.

The authors should comment about the 88% of participation rate, if it was pre-specified or not

Thank you very much. No, the 88% participation rate was not pre-specified. We have changed the sentence in the manuscript for clarification.

A General practitioner vs specialist analysis could be of interest

Thank you very much for your suggestion. That is a very interesting question: We have tested if there are differences. However, there were only minimal differences between these two groups, so we have decided not to include it in the manuscript. However, for your information, we have added these statistical evaluations for you in Table Supplemental Table 1.

Reviewer: 2

Reviewer Name: Zhixin Liu

Institution and Country: Stats Central, UNSW Sydney, Australia

Please state any competing interests or state 'None declared': None declared

Please leave your comments for the authors below

1. Statistical analysis: is there any particular reason the authors chose Fisher Exact test? Given the large sample size, Chi-square test could be more appropriate.

Thank you very much for your suggestion. We have analysed our data as we have planned and described in our study protocol, however, we have discussed this topic with our statistician. We have verified that the results of the Fisher and Chi-square test are virtually identical (p-value). Please see attached Table Supplemental 2 as one example for the comparison of Fisher vs. Chi-square test.

2. Results:

(1) Would be interested to see comparison between the internal and external group in key characteristic of physicians, e.g. age, gender, experience etc.

Thank you for your comment. We did not ask for the age of the physicians because of identification reasons. We have evaluated comparisons as gender and working experience in table 2, 4 and 5.

(2) Working experience are classified into three groups, seems the p-value reported are the overall p-value for the comparison across three groups. Further pairwise examination would help to identify where the association with the outcome is.

Thank you for your suggestion. We have also calculated the required post-hoc tests for the relevant items (where the overall test was significant). We did not add a new table in the manuscript, but added footnotes to the existing table 5. Because of the three pairwise comparisons, we also made a Bonferroni p-value correction.

Reviewer(s)' Comments to Author:

Reviewer: 1

Reviewer Name: Mario Iannaccone

Institution and Country: S.G. Bosco Hospital, Turin, Italy

Please state any competing interests or state 'None declared': none declared

Please leave your comments for the authors below

The authors should be congratulated for their efforts, the paper is interesting and original.

Thank you very much!

I have some minor comment

The authors should report in the limitation section the low response rate of the physicians as a potential bias

Thank you for your suggestion: We have added a sentence to state the potential bias of the low response rate of the physicians.

"Another important limitation of our study is the moderate response rate of external physicians that constitutes a potential bias. However, this phenomenon was also observed in other surveys with externals. It might be that external physicians can hardly be attracted for participation".

The authors should comment about the 88% of participation rate, if it was pre-specified or not

Thank you very much. No, the 88% participation rate was not pre-specified. We have changed the sentence in the manuscript for clarification.

A General practitioner vs specialist analysis could be of interest

Thank you very much for your suggestion. That is a very interesting question: We have tested if there are differences. However, there were only minimal differences between these two groups, so we have decided not to include it in the manuscript. However, for your information, we have added these statistical evaluations for you in Table Supplemental Table 1.

Reviewer: 2

Reviewer Name: Zhixin Liu

Institution and Country: Stats Central, UNSW Sydney, Australia

Please state any competing interests or state 'None declared': None declared

Please leave your comments for the authors below

1. Statistical analysis: is there any particular reason the authors chose Fisher Exact test? Given the large sample size, Chi-square test could be more appropriate.

Thank you very much for your suggestion. We have analysed our data as we have planned and described in our study protocol, however, we have discussed this topic with our statistician. We have verified that the results of the Fisher and Chi-square test are virtually identical (p-value). Please see attached Table Supplemental 2 as one example for the comparison of Fisher vs. Chi-square test.

2. Results:

(1) Would be interested to see comparison between the internal and external group in key characteristic of physicians, e.g. age, gender, experience etc.

Thank you for your comment. We did not ask for the age of the physicians because of identification reasons. We have evaluated comparisons as gender and working experience in table 2, 4 and 5.

(2) Working experience are classified into three groups, seems the p-value reported are the overall p-value for the comparison across three groups. Further pairwise examination would help to identify where the association with the outcome is.

Thank you for your suggestion. We have also calculated the required post-hoc tests for the relevant items (where the overall test was significant). We did not add a new table in the manuscript, but added footnotes to the existing table 5. Because of the three pairwise comparisons, we also made a Bonferroni p-value correction.

VERSION 2 – REVIEW

REVIEWER	Mario Iannaccone
----------	------------------

	SG. Bosco Hospita, Turin, Italy
REVIEW RETURNED	25-Nov-2019

GENERAL COMMENTS	The authors answered fully to all the raised points.
--

REVIEWER	Zhixin Liu Stats Central, UNSW Sydney, Australia
REVIEW RETURNED	19-Nov-2019

GENERAL COMMENTS	The authors sufficiently addressed my comments from previous review.
--